# Caregivers’ View of Socio-Medical Care in the Terminal Phase of Amyotrophic Lateral Sclerosis—How Can We Improve Holistic Care in ALS?

**DOI:** 10.3390/jcm11010254

**Published:** 2022-01-04

**Authors:** Katharina Linse, Elisa Aust, René Günther, Andreas Hermann

**Affiliations:** 1Deptartment of Neurology, Technische Universität Dresden, 01069 Dresden, Germany; elisa.aust@uniklinikum-dresden.de (E.A.); rene.guenther@uniklinikum-dresden.de (R.G.); 2German Center for Neurodegenerative Diseases (DZNE), Research Site Dresden, 01307 Dresden, Germany; 3Translational Neurodegeneration Section “Albrecht Kossel”, Department of Neurology, University Medical Center Rostock, 18147 Rostock, Germany; Andreas.hermann@med.uni-rostock.de; 4Center for Transdisciplinary Neurosciences, University Medical Center Rostock, 18147 Rostock, Germany; 5German Center for Neurodegenerative Diseases (DZNE), Research Site Rostock/Greifswald, 18147 Rostock, Germany

**Keywords:** amyotrophic lateral sclerosis, end-of-life care, palliative care, caregiver burden, informal caregivers

## Abstract

Multidimensional socio-medical care with an early integration of palliative principles is strongly recommended in amyotrophic lateral sclerosis (ALS), but provided inconsistently. We conducted telephone interviews with 49 former caregivers of deceased ALS patients to examine their experience of care in the terminal phase including caregiver burden. Patients who received specialized palliative care (45% of patients) were more likely to die at home (*p* = 0.004) and without burdening symptoms (*p* = 0.021). The majority of caregivers (86%) reported deficits in socio-medical care. Most frequently mentioned were problems receiving medical aids (45%) and a lack of caregiver support (35%). A higher level of deficits experienced by caregivers was associated with negative health outcomes on the side of the caregivers (reported by 57% of them; *p* = 0.002) and stronger caregiver burden (*p* = 0.004). To provide good quality of dying to patients and reduce the burden on caregivers, multidimensional—including palliative—care in ALS urgently needs to be strengthened in the healthcare structures.

## 1. Introduction

Amyotrophic lateral sclerosis (ALS) is characterized by progressive degeneration of upper and lower motor neurons. The prevalence of 3 to 5/100,000 is expected to grow in the coming decades [1]. ALS results in weakness and wasting of muscles, leading to a progressive loss of mobility and of the abilities to speak, swallow and breathe. Still, ALS is an incurable disease. Almost all ALS patients die within 2 to 5 years after diagnosis due to respiratory insufficiency, if not making use of invasive ventilation [2]. The pathophysiology of ALS is complex, which makes it unlikely to find a cure in the near future, despite recent technological advances in drug target identification and validation [3,4]. Hence, ALS-care is guided by palliative principles and aiming at the preservation of quality of life. Specialist multidisciplinary care (MDC) is strongly recommended in ALS, since care needs and care planning are complex, including life-critical decisions [5,6]. Additionally, ALS-patients report a high psychosocial load and need for support to cope with their disease. As quality of life mostly depends on patients’ psychological condition rather than on objective disease characteristics in ALS, MDC has to be holistic and focus also on individual psychological, emotional, social and spiritual needs [7,8,9], and provide access to appropriate psychosocial assistance [10,11]. MDC has been shown to improve survival, care, decision making, quality of life and patient’ satisfaction with care from diagnosis to death [12,13,14,15]. MDC must also include primary as well as specialized palliative care (PPC resp. SPC). Palliative care comprises but is not limited to all kinds of advanced care planning such as symptom management, intensive communication with patients and families, advance discussion of care preferences and end-of-life care [16,17,18]. PPC should be included rigorously from the point of diagnosis of a life-limiting diagnosis such as ALS [19]. Additionally, the provision of SPC by providers with a specific qualification and 24/7 availability is recommended to be offered to patients as soon as complex needs arise. SPC is assumed to improve distressing physical and psychological symptoms, quality of life and care planning in ALS [16,18,20,21], just as it is well known for oncological diseases [22]. Importantly, palliative and psychosocial interventions are recommended to start at time of diagnosis and to be offered continuously throughout the disease course [14,19].

It is well known that family caregivers bear the main brunt of care for ALS patients [23], and caregiver burden is high in ALS [24]. In addition to the physically and organizationally demanding tasks, caregivers are under a great deal of psychological strain: they must accept the approaching death of their loved one, endure their suffering, and live with the growing responsibilities and fears for the future [25,26,27]. Finally, widowhood after caring for a seriously ill spouse can lead to psychological distress and even complex grief disorder [28,29,30]. Psychosocial support and specialized hospice care for caregivers should therefore be an indispensable element of MDC in ALS, also beyond the patients’ death [14,26,27,31].

Despite the substantiated need and evidence-based medical guidelines that recommend MDC and the early integration of palliative care in ALS [14,32], studies from different European countries and Northern America report a limited and inconsistent provision. This includes deficits in palliative care and unmet patient and caregiver needs [8,26,33,34,35,36,37]. A recent international review concludes that ALS patients’ and caregivers’ experiences of palliative care are actually constrained to the ‘end-of-life’ stage of the illness [19]. Despite comparable needs [38], patients with ALS still receive poorer end-of-life-care than cancer patients [39].

In Germany, gravely ill or dying patients have a legal claim to SPC since 2007. However, it is unknown to which extend SPC is offered to and used by patients with ALS. In fact, care for psychosocial needs is structurally underrepresented in MDC for ALS patients, as the cost of social workers and psychologists is not routinely covered and adequate support for caregivers is not integrated into standard of care.

The aim of our study was to investigate (i) clinical characteristics and circumstances of patients dying from ALS, including integration and consequences of SPC; (ii) the family caregivers’ perception of socio-medical care and supply in the terminal phase; and (iii) burden and adverse consequences of care in informal caregivers.

## 2. Materials and Methods

The retrospective observational study was conducted at the Motoneuron disease (MND) specialist center at the Department of Neurology of the University Hospital Carl Gustav Carus of Dresden, Germany, treating MND patients from the urban area of Dresden as well as from rural regions of the federal state of Saxony. Participants were recruited using convenience sampling: family caregivers who were involved in the care for a deceased ALS patient (death 4 months to few years ago; establishes diagnosis of ALS or an ALS-variant; caregivers between 18 and 85 years old) formerly attached to the MND specialist center were invited by telephone to take part in the study. In a first telephone call, they were given details of the study and asked for their initial consent to be interviewed, which was followed up by written consent. At any time during the recruitment process and interview, participants were provided with the opportunity to ask questions about the study, interrupt or cease the interview, or skip certain questions.

Of the total of 60 caregivers contacted, 7 refused to participate on the grounds that the topic was still too emotionally stressful. Another four caregivers initially agreed to participate but were lost to follow up. Characteristics of the 49 caregivers finally participating in the study are shown in Table 1.

Interviews were conducted by telephone, using a semi-structured interview guideline designed for this study on the basis of extensive literature search and our clinical experience. Besides sociodemographic characteristics of both caregiver and patient, the interview focused on patients’ caring situation in the terminal phase and gathered information about the satisfaction or potential problems with different aspects of socio-medical care and support received by the patient, the utilization of SPC, and circumstances of dying. Additionally, duration of care, possible adverse consequences, and caregivers’ use of (psychosocial) support were assessed. Respondents’ answers were categorized using a standardized scoring system. Detailed notes were taken at any time of the interview to validate the rating.

The severity of the patient’s disabilities prior to death was quantified using the Amyotrophic Lateral Sclerosis Functional Rating Scale-Revised (ALSFRS-R; [40]). Individual scores prior death were either extracted from patients’ medical record or reconstructed with the help of the caregiver. In this validated questionnaire with a range from 0 to 48, higher scores denote better physical functioning. Additionally, background information such as date and type of onset, date of initiation of life-sustaining measures, and prescription of certain medication were obtained from the patients’ records. After completing the interview, the 10-item short version of the Burden Scale for Family Caregivers (BSFC-s; [41]) was sent to the participants. Scores in the BSFC-s range from 0 to 30 points, with higher scores indicating stronger subjective caregiver burden. Scores under 10 are classified as low, from 10 to 20 as moderate, and from 21 to 30 as severe caregiver burden.

Telephone interviews were conducted between November 2017 and November 2020 by a Master in Psychology who had been trained in conducting interviews (E.A.) and an experienced post-doctoral researcher in the field of psychological issues in MND (K.L.), who were both not part of the team consulting the patient during their lifetime. The duration of the interviews ranged from 50 to 120 min.

We used SPSS 23.0 to analyze the data. Descriptive analysis of sociodemographic data, disease characteristics, circumstances of care and dying, perceived problems in socio-medical care, and caregiver burden was carried out. Comparisons between subgroups were conducted using *t*-tests, Mann–Whitney test, ANOVA, Kruskal–Wallis test (continuous data, depending on data distribution), and chi-squared or Fisher’s exact test (categorical data). *p*-values < 0.05 were considered as statistically significant.

The study was approved by the institutional review board at the Technische Universität Dresden (EK 393122012). All subjects gave informed consent in accordance with the Declaration of Helsinki.

## 3. Results

### 3.1. Patient Characteristics

Sociodemographic and disease characteristics of the 49 deceased ALS patients are displayed in Table 2. On average, patients died 28 months after MND diagnosis, with a moderate to severe burden of symptoms, ranging up to a completely locked-in state without any possibility to communicate. Most cases of death were directly or indirectly (e.g., pneumonia) caused by respiratory insufficiency (*n* = 28; 57%); further, three patients (6%) died because of withdrawal of tracheostomy with invasive ventilation (TIV). Two patients (4%) died of a comorbid illness, while the actual cause of death was not known in 16 cases (33%).

### 3.2. Care Situation and Circumstances of Dying in Relation to SPC

The majority of patients were cared for at home (73%), 56% of them with additional support by mobile care services, and 27% lived in an inpatient care setting. SPC services were involved in 45% of cases in the terminal phase, ranging from contact with specialized outpatient palliative care services in the last three days of life up to care in a hospice for six months (see Appendix A). Compared to patients not receiving SPC, these patients died more often at home (*p* = 0.004), and none of them died in hospital. Patients receiving SPC died more often “peacefully”, and without burdening symptoms such as pain, shortness of breath, or anxiety (*p* = 0.021), all as reported by the caregivers. Additionally, patients receiving SPC spent less days in hospital in their last year of life (*p* = 0.025). Looking specifically at the patients who had expressed a wish regarding where they would like to decease (51%), almost all patients with SPC died in their preferred place (93%), while this proportion was slightly although not significantly lower in the group without SPC (70%; *p* = 0.267). Patients using a percutaneous endoscopic gastrostomy (PEG) were more likely to receive SPC than those who did not (*p* = 0.041), while this was not the case for the use of non-invasive ventilation (NIV) or TIV.

### 3.3. Deficits in Socio-Medical Care

The majority of caregivers (86%) reported at least one or more problems regarding the socio-medical care of the patients (see Figure 1). Most frequently mentioned were problems receiving medically prescribed aids (45% of cases). In 45% of the 33 cases in which professional caregivers were involved in outpatient or inpatient settings, family caregivers reported significant respective deficits. These included professional (e.g., insufficient knowledge of MND and poor care) as well as organizational shortcomings (shortage of staff and frequent changes of personnel, often associated with deployment of nurses with a lack of experience). In contrast, only one spouse reported a negative experience with SPC, stating that it was too confronting with the patient’s death.

### 3.4. Caregivers’ Situation: Health Problems and Need for Support

The family caregivers interviewed in our study provided extensive care to the patients, ranging up to round-the-clock care (see Table 1). Those reporting to be the “main caregiver” spent on average 11.4 h per day caring for the patient (range 1–18 h). Many of them reported that the patient needed help for “everything” and that they could hardly leave the house. Most caregivers were already retired (51%), but 14 (29%) had to restrict employment and one (2%) even to abandon employment due to their caregiving duties. Overall, 28 caregivers (57%) reported subjective impairment in their physical or mental health caused or exacerbated by the caregiving situation, due to overwork and/or neglect of their own health care (Figure 2). Fifteen (54%) of those caregivers reported one issue regarding their own health, nine (32%) indicated two, and four caregivers (14%) reported three such issues. Significantly more female than male caregivers reported such damages to their health (83% vs. 16%; *p* < 0.0001).

Overall, 35% of caregivers reported insufficient professional support for themselves being in the caregiving role (see Figure 1). Details regarding the received support are displayed in Table 1. Caregivers who expressed unmet needs more likely reported negative consequences of caregiving for their own health (*p* = 0.002).

### 3.5. Caregiver Burden

Caregiver burden as reported in the BSFC-s was on average on a medium level (see Table 1). Several aspects were associated with the level of caregiver burden:

#### 3.5.1. Aspects of the Caregiver

Higher caregiver burden was reported by women compared to men (14.0 ± 7.2 vs. 7.7 ± 4.5; *p* = 0.002), with a significant correlation between age and burden in women (*r* = −0.477, *p* = 0.025), while there was no effect of age in men (*r* = 0.052, *p* = 0.833). Additionally, caregivers who restricted employment reported higher caregiver burden than those who did not (14.7 ± 7.6 vs. 9.9 ± 6.2; *p* = 0.029). No significant differences in caregiver burden were found regarding family relationship (patients’ spouse vs. child; *p* = 0.584).

#### 3.5.2. Aspects of the Patient/Disease

Caregiver burden was not significantly associated with ALS severity (ALSFRS-R-score; *r* = −0.178, *p* = 0.238) or ALS duration (*r* = −0.037, *p* = 0.810), nor with the presence of PEG, NIV, or TIV (yes vs. no, respectively; *p* = 0.965; *p* = 0.957; *p* = 0.998). There were also no significant differences in caregiver burden regarding patient’s circumstances of dying (subjectively sudden vs. not sudden; *p* = 0.551; peaceful vs. with burdening symptoms; *p* = 0.731; details of circumstances known vs. unknown; *p* = 0.404).

#### 3.5.3. Aspects of the Care Situation

The higher the amount of “quality time”—defined as time that caregiver and patient spent together per day, excluding time for care activities—the lower the caregiver burden (*r* = −0.349, *p* = 0.019). In contrast, higher number of days the patient spent in hospital during the last year of their life was correlated with increased caregiver burden (*r* = 0.408, p = 0.005). No significant correlation was found between caregiver burden and the daily amount of time that caregivers spent with care (*r* = −0.142, *p* = 0.345), and caregiver burden did not differ significantly between those who stated being the “main” caregiving person compared to those who did not (*p* = 0.151). Female caregivers spent more time per week with care (50.3 ± 50.8 h vs. 40.7 ± 44.9 h, *p* = 0.428) and less quality time with the patient (20.4 ± 21.4 h vs. 24.4 ± 25.8 h; *p* = 0.727); this was associated with increased time of hospitalization during the last year (11.2 ± 14.1 vs. 5.2 ± 76; *p* = 0.086). Even though practically relevant, these differences did not reach statistical significance due to a large variance. We also observed practically relevant differences in caregiver burden between different care settings: burden was remarkably higher in caregivers of patients who were taken care of in an inpatient setting compared to those who were cared for at home, although this difference was not statistically significant (see Appendix A).

While no differences in caregiver burden were found between caregivers of patients receiving SPC and those who did not (11.0 ± 6.1 vs. 11.7 ± 7.6; *p* = 0.723), caregivers reporting more than one of the above reported deficits in socio-medical care (63%) experienced a higher burden than those reporting no or only one deficit (37%; 13.7 ± 7.3 vs. 7.4 ± 4.0; *p* = 0.002). Consistently, reporting two or more deficits was associated with a higher probability to experience health problems (*p* = 0.004) and restrictions in employment (*p* = 0.010) by the caregivers.

No association was observed between the reported caregiver burden and the time span between death and interview (*r* = −0.036; *p* = 0.813).

## 4. Discussion

Our findings, based on first-hand information from the caregivers of deceased ALS patients, send several clear messages: (i) SPC does what it is supposed to, providing a higher quality of dying in the absence of burdening symptoms and at home, if so desired, and reducing hospitalizations. (ii) There are significant deficits in socio-medical care in the terminal stage, affecting not only patients but also their caregivers. As main deficits, we identified problems with health insurance (e.g., medical aid approval), a lack of psychological and practical caregiver support, and problems with professional caregivers. (iii) Caregiver burden at least partly depends on modifiable aspects of socio-medical care in ALS. Unmet caregiver needs were associated with increased health problems of the caregivers themselves.

SPC was involved in only 45% of our cases and often only for the last weeks in life, while the inexorable nature of ALS is described as “a natural fit” for palliative care ([16], p. 137), and the scientific literature has been recommending early integration for decades [14,42]. Nevertheless, this rate was much higher than the 2.2% of severely neurologically ill patients receiving SPC in a Germany-wide data collection of the Bertelsmann Foundation [43]. One reason might be the specialization of our center and respective expertise. The value of SPC is underpinned by our results that for patients receiving SPC, informal caregivers indicated a higher quality of dying with regard to place of death and the absence of burdening symptoms. We confirm that palliative home visits can avoid unnecessary hospital admissions and days on ICU and that patients attending SPC are more likely to die in their preferred setting, which is mostly at home [44,45]. Dying at home is subjectively associated with dying peacefully, with relatively good “quality of dying” and reduced caregiver strain [45]. Accordingly, lower caregiver burden was associated with a lower amount of patients’ time in hospital in the year prior death in our sample. Objectively, dying at home is an indicator for good quality of end-of-life care, also lowering costs [33]. This is foiled by the finding that, in general, the place of death is determined by healthcare structures, but not by patients’ wishes in Germany [43]. Confirming previous studies, ALS patients likely die peacefully [46,47]. This is of great importance, as many patients and their caregivers are afraid of a painful suffocation, and this fear is associated with a wish for assisted suicide [48]. In this context, holistic palliative care has two tasks: first, to release patients and caregivers from this fear by providing information and emotional support, and second, to release or better prevent burdening symptoms by providing palliative medical support [46,48,49]. The small but relevant percentage of our patients dying with burdening symptoms illustrates that PPC alone cannot always provide such sufficient release. Typically enough, the concrete duration or amount of support by SPC could not be obtained in several cases, as patients were no longer jointly treated in the MND center as soon as they received SPC. This illustrates the lack of structural cooperation within the care system and inappropriate dichotomization between neurological and palliative medical care in practice.

Regarding socio-medical care in the terminal phase, caregivers in our study reported frequent rejection, delays, and disputes when applying for prescribed medical aids, culminating in serious constraints of patients’ autonomy and quality of life. It has been shown that medical aid provision often fails or is protracted due to unawareness of non-medically educated decision-makers [50,51,52] and is supposed to be a key problem in socio-medical ALS care [53]. This issue is of special significance, since almost all ALS patients become increasingly dependent on medical aids for mobility, communication, and daily care [50,54], and their provision is crucial for maintaining patients’ autonomy and quality of life. A timely, demand-driven supply is therefore an ethical imperative [55,56], which should prevail short-term efforts to reduce costs. Additionally, this struggle in application for medical aids drains caregivers’ energy [51,52], particularly as they generally do not receive professional support in this bureaucratic jungle.

Deficits in professional out- and inpatient care were identified as another strong stressor on caregivers, causing fears and worries and the feeling of being indispensable [52,57]. A perceived lack of non-MND center professionals’ knowledge about MND might lead caregivers to protract hospital or respite care admission and keep care in their own hands, even if this exceeds their individual load limits [31].

A subjective lack of information concerning possibilities, assets, and drawbacks of life-sustaining measures was not reported frequently, despite the complexity of this topic. In general, a timely onset of the discussion and thus decision-making process about NIV, IV, and PEG is highly recommended, particularly since patients themselves are often very insecure and tend to use a wait-and-see strategy. ACP in MND is therefore described as an individual and cyclic process in MND, which requires time, sensitivity, and expertise and is a key task of MND centers [5,6,58,59,60]. Family caregivers need to be integrated in this process, and their understanding needs to be checked, as the patient’s decisions tremendously affect their life as well [25,31]. Unduly, ACP discussions in ALS are not a billable service for neurologists in Germany. While the subjective lack of information reported by some caregivers in our sample cannot be assessed objectively, in general, a lack of information is not compatible with the aim of an autonomous, well-informed decision-making in ALS [5,55,61]. Furthermore, once the decision is made, the initiation as well as monitoring of NIV, IV, and PEG requires a high expertise about the measures taken but also about ALS. In our present sample, not all patients received optimal supply, which included delays but also insufficient support by the commercial providers. Such shortcomings may have significant consequences on survival and quality of life in ALS, up to hypoxia, avoidable hospital admissions, or premature death [62,63].

Overall, the majority of reported deficits were related to aspects of care lying outside the specialized neurologists’ sphere of influence or rather to structural problems. Accordingly, previous studies reported that caregivers and patients acknowledge the specialized medical care and expertise in MND centers, but often experience a lack of coordination of the involved actors and express a wish for counseling and assistance by a central, easily accessible contact partner [31,52,64], as also reported in our study.

In summary, the reported deficits in socio-medical care have a tremendous impact: It has been shown that unmet supportive care needs mediate the relationship between functional status and quality of life in ALS [65]. Moreover, patients’ decisions about life-sustaining measures are associated with their satisfaction with the medical system [66] and dependent from the support and guidance they receive from healthcare professionals [67].

Unfortunately, but as expected, our data furthermore underpin the current lack of sufficient specialized emotional and practical support for caregivers of ALS patients [7]. Of note, a lack of psychological support for caregivers and hurdles of bureaucracy regarding the health insurance, leading to shortcomings in the supply being the most frequently mentioned deficiencies of the medical system in another recently published German survey [53]. Looking at the consequential health damages in half of our caregiver sample and restrictions of employment in one-third, we found that caregiving caused high individual but also socioeconomic costs. This is in line with rates of health damages in consequence of caregiving in previous reports [49,53,68,69]. Importantly, medical professionals need to be aware of the fact that caregivers need to be given proactive support to prevent those consequences: often, they are trapped in their daily duties, put their own needs last, or even feel as though they have no right to respect their own load limit in the face of their relatives terminal disease and—as reported in our sample—feel not strong enough to strive for help [52,57]. Specialized psychologists—equivalent to psychooncologists—also addressing caregivers needs and worries are one urgent, maybe even the most important, need in MDC for MND patients.

Particularly noteworthy in this context is the association between caregiver burden and—potentially modifiable—aspects of the care situation, reported above. Our data clearly support the idea that caregivers and patients should be enabled to spend “quality time” together, excluding care activities, since social relationships and activities are the most valuable resource for both patient and caregiver [70,71]. This is reflected in lowered caregiver burden in our sample. This linkage is one possible explanation for the relevantly higher burden in caregivers of patients in inpatient care, where the opportunity to spend valuable time together is inevitably restricted. Another possible explanation might be the above-mentioned lack of trust in professional caregivers and accompanying feelings of guilt. Interestingly, caregiver burden was not associated with severity or duration of ALS, nor with the amount of time caregivers spent for daily care in our sample. Our finding of higher burden in female and particularly younger female caregivers might be explained by multiple role conflicts due to own employment and care for small children [72], emphasizing the need for relief for this especially vulnerable group of caregivers.

The observed association between the amount of individually experienced deficits in socio-medical care and caregiver burden, as well as adverse health and economic consequences, impressively illustrates the urgent need to address these deficits by a need-driven and evidence-based development of services. Still, previous ALS caregiver burden research mainly focusses on (not changeable) aspects of the disease and the caregivers’ personality, e.g., [24]. Meanwhile, the reduction of caregiver burden in ALS by all available means is of particular value for all parties concerned, as informal caregivers hold a double role in the disease progress: while being heavily burdened themselves, they are the most important resource for the patient and extensively involved in the care-planning process [14,25,73]. Moreover, ALS-patients’ feeling of being a burden to their families is associated with a wish towards hastened death, decisions against life-sustaining measures, and lower quality of life [60,74].

While verifying the positive effects of SPC on patients’ passing away, in accordance with a randomized controlled trial, we could not prove that caregivers themselves benefit from the involvement of this service [20]. In our study, important explanations might be the large heterogeneity of disease-related and social variables, as well as of duration and type of SPC, along with the varying amount of palliative measures provided. Additionally, a comparably short time of SPC does probably not offset the impact of longstanding caregiving. It is furthermore probable that particularly complex cases with a high physical or psychosocial load were admitted to SPC, in line with current recommendations for SPC. Moreover, it has to be taken into account that palliative interventions, especially those targeting on emotional, social, and spiritual support, might be hardly quantifiable by means of common research methods and require qualitative approaches [75]. While no effect on caregiver burden could be quantified, some caregivers emphasized that the SPC involvement was their “salvation” or psychological counseling “prevented anything worse”, referring to their psychological health, messages which are in line with the results of a qualitative study including 28 current or former caregivers of MND patients [26]. These methodic idiosyncrasies of observational palliative care research in the field of MND care might be one reason for its slow progress, which targets to identify the unmet needs towards tailored interventions and necessary improvements of healthcare for MND [8].

The most relevant limitations of our present study are the monocentric retrospective design and thus relatively small sample size. However, we suppose our data to be representative for German MND-treatment centers, as the structural, legal and financial framework of socio-medical care is similar; despite the fact, that outpatient SPC is only reimbursed by the health insurance for a limited period in some federal states. Some of the reported problems, e.g., access to psychological care or accessibility of medical aids might be even more pronounced in other countries, where health care accessibility is more dependent on the individuals’ cultural or financial background, than it is in Germany [76]. Further, decision making differs between countries: while almost 30% of ALS-patients decide for IV in Japan [77], this rate is lower in European countries, due to cultural differences and also depending on the health care structures [78], leading to differing requirements relating to MDC and SPC. Therefore, studies have to be interpreted within the concrete cultural frame and health care system. Regarding representativeness for our center, we reached a good participation rate of 82%. As however some caregivers rejected participation with the reason that the interview would be too stressful, it can be assumed that they are even more burdened and suffering from adverse consequences of caregiving in past and present. Obviously, our data mirror the caregivers’ perception, not the “objective” socio-medical care, however, subjective satisfaction is a major outcome in palliative settings. Unfortunately, we could not obtain retrospective data on patients’ potential behavioral deficits, which are a strong predictor for caregiver burden [24]. However, we assume, that these data would complement, but not change our results.

## 5. Conclusions

Given the positive impact of SPC as well as the high frequency and the potential adverse effects of deficits in socio-medical care for these vulnerable patients and their family caregivers, the results of our study once more clearly speak for the need for creating strong structures for a comprehensive MDC in ALS:

First, our data show a great need for a routinely and timely inclusion of SPC in ALS care, as it supports a better quality of dying and reduces hospitalizations. At best, this should be guided by a close collaboration between experienced MND specialists [79] and palliative actors. Regarding to this, palliative care in ALS should not be seen as a distinct medical discipline, but as a principle of care for patients with life-limiting disease, aiming at maintaining patient autonomy and patients’ as well as caregivers’ quality of life in all stages of the disease [75].

Second, deficits in socio-medical care in ALS are frequent and of a structural nature, including time-consuming approvals, especially for expensive medical aids [50]. These findings have to be consistently and systematically transferred into the healthcare system for ALS, e.g., by the inclusion of external decision-making parties such as health insurance companies. ALS care has to be understood as a holistic application of palliative principles and measures, beginning with the diagnosis, e.g., by providing optimal and timely supply with medical aids, the best professional homecare as well as emotional support. Within this frame, essential tasks such as ACP discussions as well as supportive conversations with patients as well as with caregivers should be billable. Ideally, comprehensive ALS-care should be coordinated by a fixed, easily accessible contact partner who coordinates all aspects of care throughout the whole disease course [19], e.g., a case manager. This is not only supposed to lower health costs [80], but also to provide emotional relief and a feeling of trust for patients and caregivers [64,81].

Third, burden and adverse health consequences on the part of the caregivers are associated with deficits in socio-medical care. Besides the above-mentioned elimination of these deficits, caregivers should be enabled and encouraged to spend “quality time” with the patients. Further, a regular assessment of unmet caregiver needs and proactive emotional and practical caregiver support from diagnosis beyond patients’ death should be default in ALS care.

It is essential and an ethical duty to provide the structures and concepts for a comprehensive MDC in ALS, which is based on patients’ and caregivers’ individual needs and accessible to all individuals, irrespective of their cultural, intellectual or financial background—in order to maintain at least some control in face of unremitting loss and thus promote a less traumatic experience of living and dying with the disease [19,55]

## Figures and Tables

**Figure 1 jcm-11-00254-f001:**
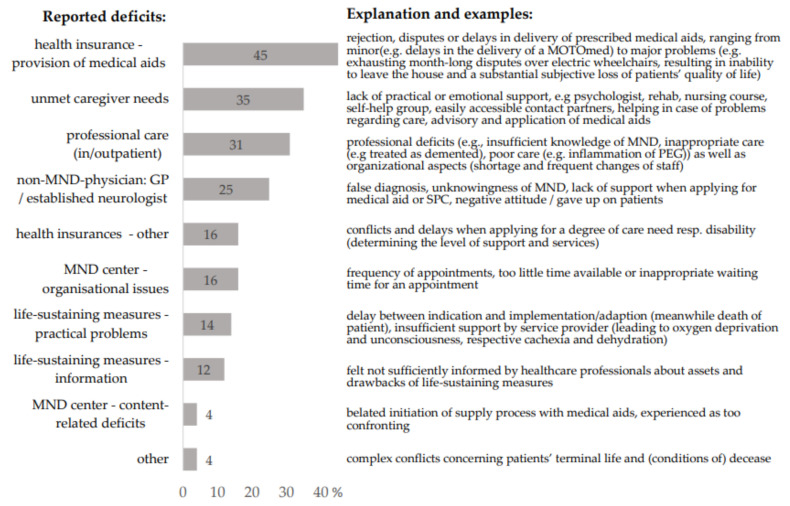
Deficits in socio-medical care reported by the caregivers. Multiple answers were possible. Data in percent. GP, general practitioner; MND, motoneuron disease.

**Figure 2 jcm-11-00254-f002:**
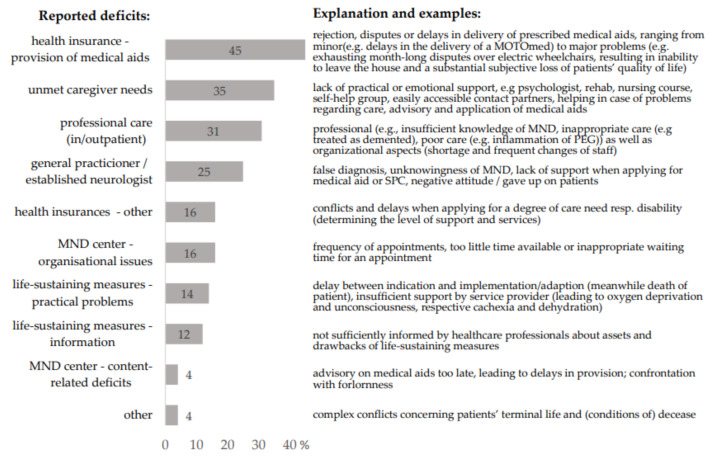
Health problems of caregivers caused or exacerbated by the caregiving situation, as reported by themselves. Data in percent.

**Table 1 jcm-11-00254-t001:** Characteristics of the *n* = 49 interviewed caregivers.

**Sociodemographics**	
Gender m:f ^a^	19:30 (39:61)
Age ^b^	63.7 ± 13.4
Education (years) ^b^	14.4 ± 2.6
**Family relationship with patient ^a^:** Spouse/partner:	33 (67)
Child:	14 (29)
Grandchild:	2 (4)
**Job status ^a^:** Retired:	25 (51)
Employed:	23 (47)
Unemployed:	1 (2)
**Impact of Caregiving**	
Shared household with patient ^a^	31 (63)
Main caregiver ^a^	27 (55)
Time spent with care (hours per week) ^b^	46.5 ± 48.4 (0–126)
Time spent together, excluding care (hours per week) ^b^	22.0 ± 23.1 (0–84)
Employment ^a^ Restricted: Abandoned:	14 (29) 1 (2)
Health problems, caused or exacerbated by caregiving ^a^	28 (57)
BSFC-s ^b^	11.4 ± 6.9 (0–29)
**BSFC-s categorical ^a^:** Low burden Moderate burden Severe burden Missing	20 (41) 23 (47) 3 (6) 3 (6)
**Provision and Lack of Professional Support**	
Psychological support provided ^a^	8 (16)
Wish for psychological support, but not provided ^a^	10 (20)
Other professional support provided, total ^a^: Unspecific offers (rehab, kinesthesia course): ALS-specific offers (peer counselling group, information event at our clinic):	9 (18) 6 (12) 4 (8)
Wish for other professional support, but not provided ^a^	15 (31)

^a^ data are presented as *n* (%); ^b^ data are presented as mean ± standard deviation (range); BSFC-s, Burden Scale for Family Caregivers short version; m:f, male:female.

**Table 2 jcm-11-00254-t002:** Patient characteristics before decease (*n* = 49).

Sociodemographics	Total *n* = 49	SPC Received *n* = 22	SPC Not Received *n* = 27	*p* *
Gender ^a^ m:f	26:23 (53:47)	10:12 (41:59)	16:11 (59:41)	0.336
Age ^b^	71.0 ± 8.5 (53–84)	69.1 ± 8.9	72.6 ± 8.0	0.190
Education (years) ^b^	14.7 ± 3.3	15.3 ± 3.9	14.2 ± 2.8	0.650
Marital status ^a^: Single: Married/partnership: Divorced: Widowed:	1 (2) 41 (84) 1 (2) 6 (12)	0 19 (86) 1 (5) 2 (9)	1 (4) 22 (81) 0 4 (15)	0.493
Care situation ^a^: Family care: Family care and nursing service: Inpatient care setting:	16 (33) 20 (41) 13 (27)	7 (32) 11 (59) 4 (23)	9 (33) 9 (33) 9 (33)	0.390
Residence ^a^: Rural: Small town (5–20T inhabs.): Medium-sized town (20–100T inhabs.): Big city (>100T inhabs.):	12 (25) 10 (20) 5 (10) 22 (45)	6 (27) 4 (18) 1 (5) 11 (50)	6 (22) 6 (22) 4 (15) 11 (41)	0.635
Time span between death and interview (months) ^b^	24.5 ± 14.9 (4–66)	23.7 ± 12.4	25.2 ± 16.9	0.723
**Disease Characteristics**				
ALSFRS-R close to death ^b^	13.9 ± 8.3 (0–30)	11.5 ± 7.4	15.8 ± 8.7	0.074
MND onset ^a^: Bulbar: Spinal: Thoracic:	13 (27) 32 (65) 4 (8)	6 (27) 14 (64) 2 (9)	7 (26) 18 (67) 2 (7)	0.967
MND subtype ^a^: ALS: ALS-FTD: PMA: Completely locked in syndrome:	39 (80) 2 (4) 5 (10) 3 (6)	17 (77) 2 (9) 1 (5) 2 (9)	22 (81) 0 4 (15) 1 (4)	0.156
Age at diagnosis (years) ^b^	68.6 ± 9.5 (44–83)	66.1 ± 10.5	70.7 ± 8.3	0.138
Duration MND diagnosis to death (months) ^b^	28.1 ± 29.6 (1–162)	34.6 ± 38.2	23.0 ± 19.3	0.350
PEG ^b^,duration (months) ^b^	19 (39); 18.2 ± 24.4 (1–81)	12 (55); 21.8 ± 25.9	7 (26); 12.0 ± 22.1	**0.041**; 0.115
NIV ^b^,duration (months) ^b^	16 (33); 10.9 ± 16.1 (1–68)	5 (23); 8.6 ± 2.2	11 (41); 12.0 ± 19.6	0.181; 0.154
TIV ^b^,duration (months) ^b^	8 (16); 32.3 ± 31.3 (1–81)	4 (18); 47.5 ± 30.8	4 (14); 17.0 ± 26.7	1.00; 0.185
**Circumstances of Dying**				
Place of death ^a^: At home: Nursing home: Hospital, emergency case: Hospital, planned admission: Hospice/palliative care unit:	24 (49) 11 (22) 7 (14) 3 (6) 4 (8)	15 (68) 3 (14) 0 0 4 (18)	9 (33) 8 (30) 7 (26) 3 (11) 0	**0.004**
Death subjectively “sudden”, as perceived by fCG ^a^	29 (59)	9 (41)	20 (74)	**0.028**
Ratio: Died in preferred place/expressed wish for preferred place of death ^a^	21/25 (84%)	14/15 (93%)	7/10 (70%)	0.267
Dying ^a^: Peacefully: With burdening symptoms (pain, fear, agitation, dyspnea):	37 (76) 7 (14)	22 (100) 0	15 (56) 7 (26)	**0.021**
Exact circumstances of dying unknown to fCG ^a^	11 (22)	1 (5)	10 (37)	**0.013**
Relieving medication used in dying phase ^a^	30 (61)	18 (82)	12 (44)	0.104
Days in hospital in last 12 months of life ^a^	8.9 ± 12.4 (0–63)	4.9 ± 7.8	12.2 ± 14.5	**0.025**
**Psychological Impact**				
Wish for physician-assisted suicide by patient, as reported by fCG ^a^: No: Yes: Undecided: Unknown:				0.292
	23 (47)	11 (50)	12 (44)	
	12 (25)	7 (32)	5 (19)	
	2 (4)	1 (5)	1 (4)	
	12 (24)	3 (14)	9 (33)	
BSFC-s of fCG ^b^ (*n* = 46)	11.4 ± 6.9 (0–29)	11.0 ± 6.1	11.7 ± 7.6	0.723

^a^ data are presented as *n* (%); ^b^ data are presented as mean ± standard deviation; * *t*-test, Mann–Whitney test, chi^2^ test, or Fisher’s exact test, as appropriate; ALSFRS-R, Amyotrophic Lateral Sclerosis Functional Rating Scale-Revised; ALS-FTD, Amyotrophic lateral sclerosis with frontotemporal dementia; BSFC-s, Burden Scale for Family Caregivers short version; fCG, family caregiver; MND, motoneuron disease; NIV, non-invasive ventilation; PEG, percutaneous endoscopic gastrostomy; PMA, progressive muscular atrophy; SPC, specialized palliative care; TIV, invasive ventilation with tracheostomy; The bold means *p*-values reached significance.

## Data Availability

Data available on request.

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
