# Peer review of "Caregivers’ View of Socio-Medical Care in the Terminal Phase of Amyotrophic Lateral Sclerosis—How Can We Improve Holistic Care in ALS?"

_jcm, 2022, doi:10.3390/jcm11010254_

Round 1

Reviewer 1 Report

Dear Editor,

The manuscript by Linse et al. conducted a study that involves telephone interviews with 49 former caregivers of deceased ALS patients to examine their experience of care in the terminal phase including caregiver burden.

The design of the study and the technical quality of the work look convincing and results can be of general interest. The manuscript is well-written and easy to follow. 

Best.

Minor:

-ALS is yet an incurable disease. Part of the problem is the complexity of pathophysiology that underlines ALS and other neurodegenerative conditions. This means finding a cure won’t be expected to happen in the near future despite the recent advances in target identification and validation. Hence the implementation of palliative-care in ALS urgently needs to be strengthened in the health care structures. This should be discussed early in the introduction to provide a stronger foundation for the importance of this work. References:

https://pubmed.ncbi.nlm.nih.gov/33672148/

https://pubmed.ncbi.nlm.nih.gov/33925236/

Author Response

Dear reviewer,

we deeply thank you for your positive evaluation of our study and for you helpful comment on the discussion.

Comment: ALS is yet an incurable disease. Part of the problem is the complexity of pathophysiology that underlines ALS and other neurodegenerative conditions. This means finding a cure won’t be expected to happen in the near future despite the recent advances in target identification and validation. Hence the implementation of palliative-care in ALS urgently needs to be strengthened in the health care structures. This should be discussed early in the introduction to provide a stronger foundation for the importance of this work.

Response: We changed this as indicated in the revised manuscript and integrated the suggested literature.

By following the comments of both reviewers we believe that our revised manuscript significantly improved.

Yours sincerly,

Katharina Linse 

Reviewer 2 Report

Dr. Linse and colleagues presented an instructive experience study on caregivers of ALS patients in this manuscript.  The authors interviewed 49 informal caregivers and gathered data on a variety of factors. The manuscript is well written, and the conclusion will help to enhance ALS patient care guidelines, particularly for local patients. The main limitation for this study, also as the author mentioned, is its monocentric retrospective design and small sample size. Given that this study was conducted in only one location in Germany, I would suggest the authors to add discussions on the impact of culture, economies and health care systems that varied by countries or regions.

Author Response

Dear reviewer,

we deeply thank you for your positive evaluation of our study and for you helpful comment on the discussion.

Comment: I would suggest the authors to add discussions on the impact of culture, economies and health care systems that varied by countries or regions.

Response: We changed this as indicated in the revised manuscript.

By following the comments of both reviewers we believe that our revised manuscript significantly improved.

Yours sincerly,

Katharina Linse